# A Traumatic Neuroma Formation Following Fasciotomy for the Treatment of Tibialis Anterior Muscle Herniation: A Case Report

**DOI:** 10.3390/medicina59030466

**Published:** 2023-02-27

**Authors:** Takuji Yokoe, Takuya Tajima, Nami Yamaguchi, Yudai Morita, Etsuo Chosa

**Affiliations:** Division of Orthopaedic Surgery, Department of Medicine of Sensory and Motor Organs, Faculty of Medicine, University of Miyazaki, 5200 Kihara, Kiyotake, Miyazaki 889-1692, Japan

**Keywords:** fasciotomy, hernia, muscle, skeletal, neuroma, peroneal nerve

## Abstract

Muscle herniation of the lower extremity, such as tibialis anterior muscle herniation (TAMH), is not a rare cause of leg pain in athletes. However, a few studies have reported surgical treatment for TAMH, and the optimal surgical procedure remains controversial. Fasciotomy was reported to be effective for patients with TAMH. However, this procedure would be associated with a risk of intraoperative injury to the superficial peroneal nerve (SPN), although no previous literature has reported this complication. This case report aimed to report a case of bilateral TAMHs in which a traumatic neuroma of the SPN developed after fasciotomy. A 16-year-old baseball player presented with painful swelling lesions of the bilateral lower extremities (1 lesion on the right, 3 lesions on the left) after sports activities. An ultrasonographic evaluation showed swelling lesions of the anterolateral parts of the bilateral lower extremities in the standing position after dashing, while these lesions were not detected in the supine position. A fasciotomy of the crural fascia was performed after conservative treatment failed. Several days after surgery, the patient presented with weakened touch sensation over the dorsal area of the left foot. At the three-month follow-up examination, a swelling lesion with hard elasticity was identified. The palpation of this lesion caused a radiating sensation in the area supplied by the SPN. He was able to return to playing baseball six months after surgery. The patient was asymptomatic without palpation of the traumatic neuroma of the SPN at the latest follow-up examination. In conclusion, the present case report suggests that orthopedic surgeons need to consider the risk of iatrogenic injury to the SPN during fasciotomy for the treatment of TAMHs. However, there may be a risk of injuring the SPN because of the many variants of the course of the SPN within the compartment of the lower extremities.

## 1. Introduction

Muscle herniation of the lower extremity is reported to occur after blunt traumatic injury to the lower leg or muscle hypertrophy due to strenuous activities in athletes or soldiers [1,2]. Lee et al. reported that tibialis anterior muscle herniation (TAMH) was the most common site of muscle herniation [3]. When conservative treatments do not resolve the symptoms, surgical intervention is generally required. Available surgical options previously reported in academic papers include direct repair [4], repair using periosteum [5] or mesh [6,7] and fasciotomy [2,8]. These previous studies reported favorable clinical outcomes. However, the majority of previous studies were retrospective case series, and the best surgical procedure for TAMH remains a matter of debate. Fasciotomy was reported to be safe and provide favorable outcomes for the treatment of symptomatic muscle herniation of the lower extremities [8]. However, no studies have evaluated the optimal length of fasciotomy or the preferred device for performing fasciotomy to TAMH treatment.

When fasciotomy is performed for the treatment of TAMH, similar to that for the treatment of chronic exertional compartment syndrome (CECS) of the lower leg, clinicians should pay attention to iatrogenic damage to the superficial peroneal nerve (SPN) [9,10,11,12]. Grechenig et al. reported a minimally invasive technique using two incisions as a promising strategy to decrease the risk of iatrogenic injury of the SPN for patients with CECS [10]. Additionally, several anatomical studies have shown that there are many variants of the course to the SPN and its branches within the anterior compartment of the lower extremities [13,14]. The location of the TAMH may be associated with an increased risk of iatrogenic injury to the SPN when performing a fasciotomy. However, to our knowledge, no studies have reported or discussed the risk of iatrogenic injury to the SPN during the surgical treatment of TAMHs. To perform fasciotomies effectively and safely for patients with TAMHs, recognizing the risk of iatrogenic injury to the SPN is crucial.

This case report aimed to show a case of bilateral multiple TAMHs in which a traumatic neuroma of the SPN developed following fasciotomy. Informed consent was obtained from the patient and his parents for the publication of this report and any accompanying images.

## 2. Case Presentation

A 16-year-old baseball player was referred to our hospital for the assessment of multiple swelling lesions in the bilateral anterolateral lower legs. He had suffered from painful lesions when playing baseball for about a year, limiting his athletic performance. He had no history of traumatic injuries to the lower legs. Clinical examinations showed bilateral painful swelling lesions (1 on the right and 3 on the left) after dashing (Figure 1). These lesions were concave on palpation at rest. Ultrasonographically, these lesions did not swell with the patient in the supine position but were prominent in the standing position following dashing (Figure 2). Magnetic resonance imaging (MRI) showed a thin or void crural fascia (Figure 3). The preoperative visual analog scale (VAS) was 6/10 when the lesions were symptomatic. The patient was diagnosed with bilateral TAMHs.

Fasciotomy of the crural fascia was performed after two months of conservative treatment failed. The surgery was performed under spinal anesthesia with the patient in the supine position. A tourniquet was used for exsanguination. Firstly, a 30-mm longitudinal incision was made over the lesion in the right lower leg. When the crural fascia was exposed, thin fascia was detected (Figure 4A). A fasciotomy of about 60 mm length was performed. It was subsequently confirmed that the tibialis anterior muscle did not herniate from repetitive dorsiflexion of the ankle. Thereafter, fasciotomies were performed for the left lesions. On the left side, there were three separate lesions. Therefore, three longitudinal incisions were separately made over each lesion (Figure 4B,C). All lesions had thin crural fascia. After confirming that the SPN was not visible, a fasciotomy was performed using a subcutaneous tunnel. The wounds were closed in a standard fashion.

Postoperatively, the ankles were not immobilized, and weight bearing was allowed as tolerated. Ankle range-of-motion exercises were initiated immediately after surgery. Several days after surgery, he presented with paresthesia and impaired touch sensation over the dorsal area of the left foot. At the three-month follow-up, a swelling lesion with hard elasticity was identified between the middle and the most distal surgical wounds. The lesion did not disappear with the patient in the supine position. On palpating this lesion, he recognized a radiating sensation over the left dorsal foot supplied by the SPN. An ultrasonographic examination showed an isolated lesion (8.0 × 12.0 mm) (Figure 5). MRI showed a lesion that was iso-signal on T1-weighted images and high signal on T2-weighted images (Figure 6). Traumatic neuroma of the SPN was considered. While the lesion did not disappear, he returned to playing baseball six months after surgery. At the 24-month follow-up examination, recurrent TAMHs were not found in the bilateral lower extremities. The patient was asymptomatic without palpation of the lesion and did not demand surgical intervention for this lesion.

## 3. Discussion

First, we presented a case of bilateral multiple TAMHs with traumatic neuromas of the SPN following fasciotomy. Several surgical procedures have been reported for the treatment of TAMH [2,4,5,6,7,8,15]. However, these previous studies were case series without high-level evidence, indicating no established surgical treatment for TAMHs. Kramer et al. reported that fasciotomy was safe and effective for young athletes presenting with symptomatic muscle herniations of the lower extremities [8], which is the largest case series that evaluated postoperative outcomes following fasciotomy for patients with muscle herniation of the lower extremities. In the present case, bilateral TAMHs were identified. Furthermore, three lesions were present in the left lower extremity. Considering that repair using mesh or periosteum would not be appropriate for these multiple TAMHs due to the medical cost and number of available autografts, fasciotomy was selected.

In the present case, traumatic neuroma of the SPN occurred after fasciotomy. Several studies have demonstrated that there are many variants of the course of the SPN and its branches within the compartment of the lower extremities [13,14,16,17]. Apaydin et al. described three types of courses of the SPN: type 1 (71%), where the SPN runs entirely within the lateral compartment; type 2 (23.7%), where the SPN runs within the anterior compartment below 12.7 cm from the tip of the fibular head; and type 3 (5.3%), where the branches of the SPN run both in the anterior and lateral compartments [13]. In the present case, fasciotomy was performed after confirming that the SPN was not detectable over the crural fascia intraoperatively. Therefore, it was speculated that the patient had a type 2 or 3 SPN course, according to the classification by Apaydin et al. It was reported that blind subcutaneous fasciotomies caused traumatic damage to the SPN in 12.5%–18.7% of cases [18]. To avoid iatrogenic injury to the SPN during fasciotomy for CECS, an anatomical investigation of the course of the SPN has been performed by several authors [9,10,11,12]. Some authors have introduced arthroscopy-assisted fasciotomy as a promising technique for reducing the risk of injury to the SPN [19,20,21]. Grechenig et al. reported a minimally invasive technique using 2 incisions, with an incidence of injury to the SPN of 5% (2/40 cadaveric specimens) [10]. Arthroscopy-assisted fasciotomy or the two-incision technique may decrease the risk of iatrogenic injury to the SPN during fasciotomy. However, these techniques are not completely free from the risk of injury to the SPN [10,11,19]. Additionally, these techniques are not effective for decreasing the risk of injury to the SPN in type 2 or 3 courses [13], as the SPN or its branches run under the crural fascia of the anterior compartment in such situations. Therefore, surgeons need to consider possible variants in the SPN course and be aware of the risk of iatrogenic injury to the SPN regardless of the technique used during fasciotomy for the treatment of TAMHs, similar to cases of CECS.

## 4. Conclusions

We presented a case of bilateral multiple TAMHs in which a traumatic neuroma of the SPN developed after fasciotomy. For the treatment of TAMHs, clinicians need to consider the risk of iatrogenic injury to the SPN during fasciotomy.

## Figures and Tables

**Figure 1 medicina-59-00466-f001:**
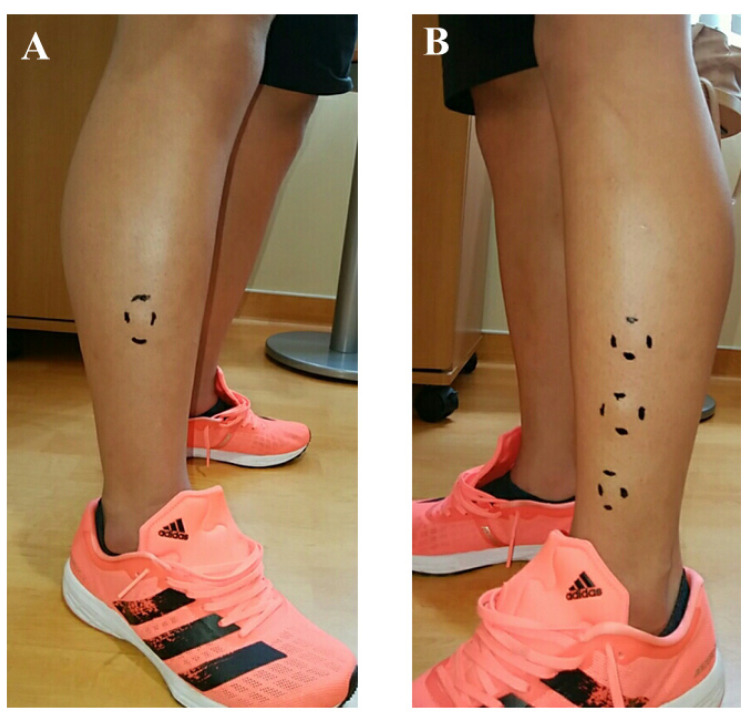
Preoperative photographs of the bilateral lower extremities after strenuous activity. (**A**) A swelling lesion was present in the right lower leg. (**B**) Three swelling lesions were present in the left lower leg.

**Figure 2 medicina-59-00466-f002:**
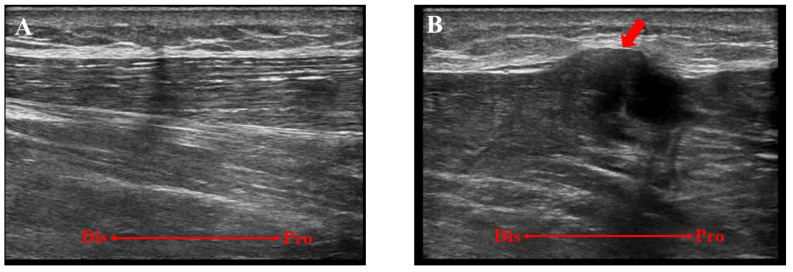
A preoperative ultrasonographic evaluation of the left lower extremity. (**A**) The lesion was not detectable with the patient in the supine position. (**B**) The lesion (red arrow) was swollen and detectable in the standing position after strenuous activity. Dis, distal; Pro, proximal.

**Figure 3 medicina-59-00466-f003:**
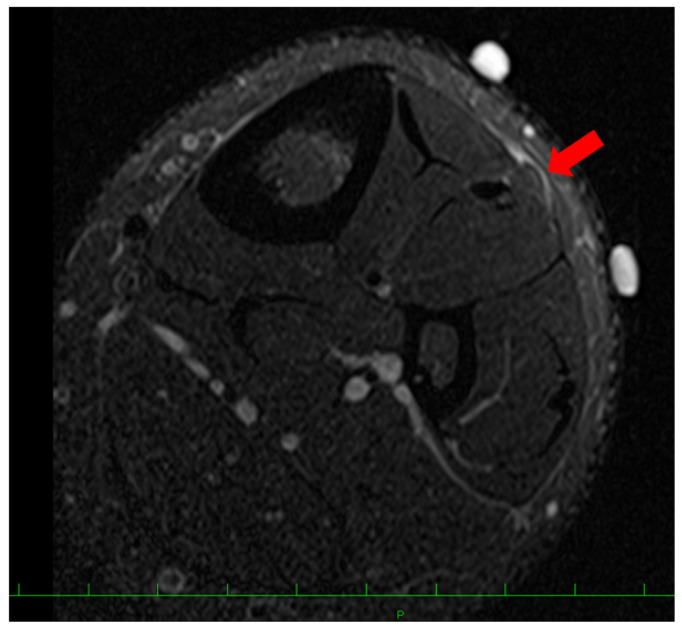
Preoperative magnetic resonance imaging (MRI) findings. A slightly swollen lesion (red arrow) of the tibialis anterior muscle with thin crural fascia was detected.

**Figure 4 medicina-59-00466-f004:**
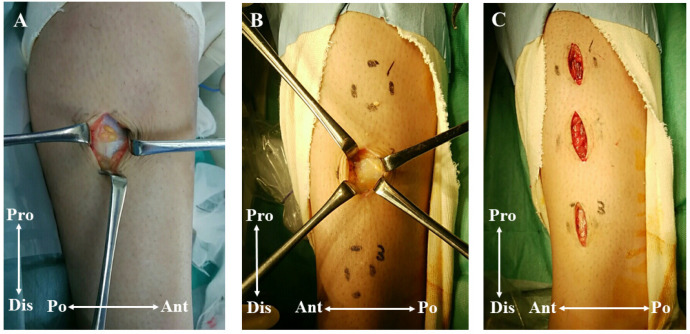
Intraoperative photographs. (**A**) The right lower extremity before fasciotomy. The crural fascia of the lesion was very thin. (**B**) The left lower extremity before fasciotomy. The crural fascia of the lesion was also very thin. (**C**) The left lower extremity after fasciotomy. Surgical incisions were made just over each lesion. Pro, proximal; Dis, distal; Po, posterior; Ant, anterior.

**Figure 5 medicina-59-00466-f005:**
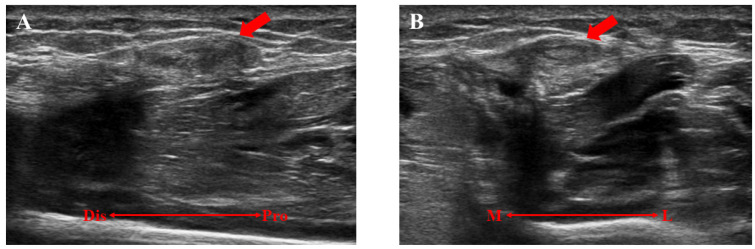
The ultrasonographic findings of the traumatic neuroma of the superficial peroneal nerve (12 × 8 mm) (red arrow). (**A**) A long-axis view. (**B**) A short-axis view. Dis, distal; Pro, proximal; M, medial; L, lateral.

**Figure 6 medicina-59-00466-f006:**
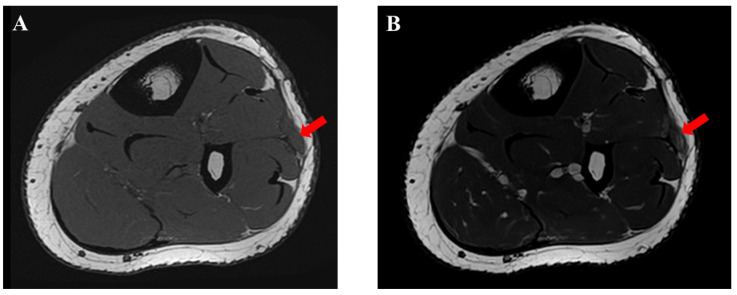
Postoperative magnetic resonance imaging (MRI) findings of the traumatic neuroma of the superficial peroneal nerve (red arrow). (**A**) T1-weighted image. (**B**) T2-weighted image.

## Data Availability

The data presented in the present study are available upon request from the corresponding author.

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
