# Peer review of "A Traumatic Neuroma Formation Following Fasciotomy for the Treatment of Tibialis Anterior Muscle Herniation: A Case Report"

_medicina, 2023, doi:10.3390/medicina59030466_

Round 1

Reviewer 1 Report (Previous Reviewer 1)

I thank the corresponding author for their comments. I have read through the subsequent changes made to the manuscript and I have no further comments or suggestions.

Reviewer 2 Report (Previous Reviewer 2)

I do not have any other comments.

This manuscript is a resubmission of an earlier submission. The following is a list of the peer review reports and author responses from that submission.

Round 1

Reviewer 1 Report

Originality:  This is a very paper, providing a maximum of information about a case study, and constitutes a very important contribution to the literature.  I think it would be a more clear case report if the following parts were revised and supplemented. These will be discussed below relative to the information of the manuscript.

Specific comments:

Title: The title of this manuscript is very long. Perhaps a more concise version for clarity, interes and ease of read.

Abstract: It is hard to get the detail in an abstract when the word count is limited and this is often the hardest part of a paper to write. However, I do feel that it would be beneficial to explain the aim and conclusions what specifically you are looking at in relation to outcomes in this case report. This needs to be made clearer throughout the paper

Keywords: Please use recognised MeSH terms as this will assist others when they are searching for information on your research topic. The following website will provide these (simply start typing in a keyword and see if it exists or find an alternative if it does not): https://www.ncbi.nlm.nih.gov/mesh

The introduction is weak and very short. An introduction should announce your topic, provide context and a rationale for your work, while catching the reader´s interest and attention. The above has not been given in the introduction that I have read.

Methodologically Sound:  As a case study report it is rather hard to go wrong methodologically, and the paper conforms to the standard.

Follows Appropriate Ethical Guidelines: Whilst there is no obvious declaration of ethical approval. Please include the date and code register number of ethics committee

 it would appear to be a report of actions taken as part of normal clinical practice (as a case study report), and thus is acceptable. 

Has results which are clearly presented and support the conclusions: Again, it conforms to the usual format for the presentation of a case study, although the content is very long.  It is, however, appropriate enough, and does report a rare case likely to be of interest to a healthcare audience.  

Overall Scientific Quality:  As a case study report it have scientific depth, but effectIvely is intended only to report the occurrence of a typical case and to highlight the importance of correct diagnosis, and on these grounds merits attention. 

Presentation, Organization, Clarity:  I think you have some good information. 

Correctly References Previous Relevant Work:  It appears to reference prior work succinctly and accurately. 

Importance/Interest: Although marked by its brevity, the content is of interest, particularly to clinicians and nursing  who examine syndrome a great deal of the time, and physiotherapists  who may need to be aware of the variant forms of this illness.  

Reviewer 2 Report

1.      The abstract requires the addition of quantitative results.

2.      Given the “take-home” message at the end of the abstract, the present form was insufficient.

3.      Reorder keywords based on alphabetical order.

4.      Nothing truly unique in its current state. Because of the lack of novel, the current study looks to be a replication or modified study. The authors must describe their novel in detail. This work should be rejected owing to a major issue.

5.      It is essential to summarize previous studies' merits, novelties, and limitations in the introductory part to emphasize the gaps in the research that the latest research seeks to address.

6.      Line 43, please do not use “we”, make it into passive for more scientifically sound.

7.      To help the reader grasp the study's workflow more easily, the authors could include more visuals to the materials and methods section in the form of figures rather than sticking with the text that now predominates.

8.      What is the baseline of patient selection? Is there any protocol, standard, or basis that has been followed? It is unclear since the patient is very heterogeneous with a small number. The resonance involved impacts the present result makes this study flaws. One major reason for rejecting this paper.

9.      The authors encouraged to discuss potential further study via computational simulation that offer main advantages rather than clinical study as conducted in the present study such as lower cost and faster results. Also additional relevant reference is needed as follows: Minimizing Risk of Failure from Ceramic-on-Ceramic Total Hip Prosthesis by Selecting Ceramic Materials Based on Tresca Stress. Sustain. 2022, 14, 1–12. https://doi.org/10.3390/su142013413

10.   It is necessary to provide more information on the manufacturer, country, and specifications of the tools.

11.   Error and tolerance of experimental tools used in this work are important information that needs to be explained in the manuscript. It is would use as a valuable discussion due to different results in the further study by other researcher.

12.   A comparative assessment with similar previous research is required.

13.   The discussion in present article is extremely poor in quality as overall. The authors must elaborate on their arguments and provide a thorough justification. Don't just state the results and give a quick explanation.

14.   Before moving on to the conclusion section, the present study's limitation must be added at end of the discussion section.

15.   Mention further research in the conclusion section.